# The Association between Fast Food Outlets and Overweight in Adolescents Is Confounded by Neighbourhood Deprivation: A Longitudinal Analysis of the Millennium Cohort Study

**DOI:** 10.3390/ijerph182413212

**Published:** 2021-12-15

**Authors:** Mark A. Green, Matthew Hobbs, Ding Ding, Michael Widener, John Murray, Lindsey Reece, Alex Singleton

**Affiliations:** 1Geographic Data Science Lab, Department of Geography & Planning, University of Liverpool, Liverpool L69 7ZT, UK; jcmurray@liverpool.ac.uk (J.M.); ucfnale@liverpool.ac.uk (A.S.); 2GeoHealth Laboratory, University of Canterbury, Christchurch 8140, New Zealand; matt.hobbs@canterbury.ac.nz; 3School of Health Sciences, University of Canterbury, Christchurch 8140, New Zealand; 4School of Public Health, University of Sydney, Sydney 2006, Australia; melody.ding@sydney.edu.au (D.D.); lindsey.reece@sydney.edu.au (L.R.); 5Department of Geography & Planning, University of Toronto, Toronto, ON M5S 3G3, Canada; michael.widener@utoronto.ca

**Keywords:** fast food, neighbourhood, deprivation, overweight, obesity, adolescence, confounding

## Abstract

The aim of our study is to utilise longitudinal data to explore if the association between the retail fast food environment and overweight in adolescents is confounded by neighbourhood deprivation. Data from the Millennium Cohort Study for England were obtained for waves 5 (ages 11/12; 2011/12; *n* = 13,469) and 6 (ages 14/15; 2014/15; *n* = 11,884). Our outcome variable was overweight/obesity defined using age and sex-specific International Obesity Task Force cut points. Individuals were linked, based on their residential location, to data on the density of fast food outlets and neighbourhood deprivation. Structural Equation Models were used to model associations and test for observed confounding. A small positive association was initially detected between fast food outlets and overweight (e.g., at age 11/12, Odds Ratio (OR) = 1.0006, 95% Confidence Intervals (CI) = 1.0002–1.0009). Following adjusting for the confounding role of neighbourhood deprivation, this association was non-significant. Individuals who resided in the most deprived neighbourhoods had higher odds of overweight than individuals in the least deprived neighbourhoods (e.g., at age 11/12 OR = 1.95, 95% CIs = 1.64–2.32). Neighbourhood deprivation was also positively associated to the density of fast food outlets (at age 11/12 Incidence Rate Ratio = 3.03, 95% CIs = 2.80–3.28).

## 1. Introduction

Socio-ecological models have long emphasised the importance of environmental factors in understanding obesity [1]. One prominent area of investigation focuses on the (retail) fast food environment (FFE), defined here as shops and restaurants in neighbourhoods with a high speed of service selling energy dense and nutritionally poor foods (commonly referred to as ‘fast food’). Studies have found associations demonstrating that individuals who live in areas with a greater number (density) of fast food outlets had larger body weights [2,3,4]. Possible explanations for these associations include fast food being more readily available or easier to access out of the home (and conversely fewer outlets selling healthier foods), and outlets acting as visual advertisements that nudge people’s dietary choices later at home. Such evidence has underpinned efforts, for instance in the UK [5], Canada [6], and Australia [7], by policymakers to tackle fast food environments through utilising planning regulations to limit the locations of new fast food outlets (e.g., surrounding schools or in areas of high density of existing outlets). Adolescent health is often the focus of such interventions, partly due to adolescents’ high consumption of fast food and since dietary behaviours can become habitual into adulthood [8]. While dietary choices are often determined by parents, secondary school children can often leave school premises during lunch or may access outlets when walking home [5].

The translation of evidence into policy action may be misguided when viewed alongside the whole context of the evidence base. Systematic reviews have persistently demonstrated inconsistent associations between measures of the FFE (e.g., density of fast food outlets) and obesity-related outcomes [9,10,11,12]. One recent systematic review concluded that null associations dominated findings, comprising 76.0% of the 1937 associations analysed [13]. There are several possible reasons for the inconsistency in associations across the literature. First, there may be no association and the smaller proportion of studies that do find a positive association are actually spurious associations. Second, inconsistent study design and methods, and a lack of transparency in reporting of decisions, make it difficult to compare findings [13,14,15,16]. It is plausible that those reporting positive associations may reflect either effective or poorly designed studies. Third, cross-sectional studies dominate the literature and are often less suitable for identifying relationships, especially as longitudinal studies tend to find null associations [17,18]. Fourth, a focus on local case studies in specific cities or regions may produce results that are less generalisable elsewhere or to national populations. Fifth, analyses are often purely associational rather than testing specific pathways or mechanisms for how and why the FFE matters [19,20,21]. 

One additional explanation for the existence of positive associations among a larger number of null findings often not considered may relate to the mechanistic role of neighbourhood deprivation. Neighbourhood deprivation is strongly associated with both obesity-related outcomes and the locations of fast food outlets. Individuals who reside in poorer areas are more likely to be obese, due to the complex interplay between materialistic (e.g., fewer resources to afford a healthy diet), psychosocial (e.g., lower sense of control) and geographical (e.g., poor access to healthy food outlets) factors disadvantaged people often face [22,23,24]. Fast food outlets tend to cluster in deprived areas due to cheaper rents and greater social desirability [25,26]. As such, this dual relationship suggests that neighbourhood deprivation may confound the association between the FFE and obesity-related outcomes. Separating out the independent effects of neighbourhood deprivation from fast-food outlets is therefore difficult [20]. Most studies simply adjust for neighbourhood deprivation within a regression model. However, this is not always appropriate, as it only examines the association to the outcome variable rather than adjusting for the multiple relationships and pathways through how exposure and control variables may be inter-related (including any observed confounding pathways), and a residual effect may be left through strongly correlated variables [27]. Accounting for the correct pathway for how neighbourhood deprivation operates is key to robustly assessing if the FFE matters for obesity-related outcomes. We are not aware of any previous research that has rigorously evaluated the confounding nature of neighbourhood deprivation on obesity-related outcomes in adolescents.

The aim of our study is to utilise longitudinal and representative national data to explore the extent that the association between the fast food environment and overweight in adolescents is confounded by neighbourhood deprivation. To help guide our investigation, we define the following hypotheses:

1. Individuals who live in areas with more fast food outlets have a higher likelihood of being overweight—Here we hypothesise there is an association between FFE and body weight when just considering these two variables alone. Our hypothesis follows evidence in the literature of a positive association between density of fast food outlets and measures of body weight [2,3,4].

2. Individuals who live in deprived neighbourhoods have a higher likelihood of being overweight—We hypothesise that neighbourhood deprivation is related to body weight when no other variables are considered. This follows evidence in the literature demonstrating that poor social disadvantage is associated with obesity-related outcomes [22,23,24]. 

3. Neighbourhood deprivation confounds the association between fast food outlets and likelihood of being overweight—When we explicitly account for the proposed confounding nature of neighbourhood deprivation on the FFE, we find that the association between the FFE and overweight disappears. Similarly, we expect to find a positive relationship between deprivation and the FFE, and deprivation and overweight. We expect this due to the dual relationship neighbourhood deprivation has with both obesity-related outcomes [22,23,24] and the locations of fast food outlets [25,26].

4. In areas where the number of fast food outlets increased, individuals were not more likely to be overweight—Using a quasi-experimental longitudinal design to our study [28], we examine if changes in our exposures (i.e., increasing or decreasing numbers of fast food outlets over time) were associated with changes in our outcome (i.e., risk of overweight). We hypothesise that because there is no real association between FFE and overweight, individuals who moved to areas with more fast food outlets were not more likely to be overweight than those who moved elsewhere. There are relatively few studies that have explored the temporal element of these associations, partly due to the dominance of cross-sectional studies [11,18]. 

5. The confounding role of neighbourhood deprivation persists even after controlling for diet and physical activity—We next adjust our model to incorporate two key determinants of body weight that may sit on the pathways (e.g., as mediators) between associations for density of fast food outlets and neighbourhood deprivation to risk of overweight; consumption of fast food and physical activity. Specifically, we hypothesise that (i) individuals who consume fast food or are physically inactive are more likely to be overweight, (ii) individuals who are exposed to more fast food outlets are more likely to consume fast food, and (iii) individuals from deprived neighbourhoods are more likely to consume fast food and be physically inactive. These hypotheses are grounded in existing research, especially for understanding social inequalities in health [24].

## 2. Materials and Methods

### 2.1. Participants and Setting 

The Millennium Cohort Study (MCS) is a UK representative longitudinal cohort survey following the lives of 18,827 children born in 2000 (Connelly and Platt, 2014). Waves 5 (ages 11/12; 2011/12; *n* = 13,469) and 6 (ages 14/15; 2014/15; *n* = 11,884) were used to match the same period of food outlet data. These were the only waves that we had fast food location data for at the time of analysis. Participants who resided in England in both waves (*n* = 9736) were selected for our analytical sample to match our FFE measures. Special access to the Lower Super Output Area (LSOA) codes of participants in MCS were granted by the data controllers. LSOAs represent statistical zones equivalent to small neighbourhoods (~1500 people) that are commonly used to match individuals to small area data (with smaller zones not available in this analysis). Full residential addresses or postcodes were not available, meaning we had to rely on area summaries (rather than defining areas around houses) which may produce less accurate measures of our exposures (especially for large areas).

### 2.2. Outcome: Overweight

Interviewers objectively measured anthropometrics including height and weight of participants. Overweight (including obesity) was defined using age and sex-specific International Obesity Task Force (IOTF) cut points [29]. As our outcome was adjusted for age and sex, they were not accounted for as covariates in the analysis. Overweight and obesity were considered together since (i) this is a common outcome variable when studying adolescents and (ii) excess body weight in adolescents is associated with obesity in later life [8], as well as current and future health [30]. While the use of a categorical variable is limited through hiding variations within groups, z-score body weight values (and height and weight) were not provided by the data controller for our analysis due to their high sensitivity. 

### 2.3. Exposure: Fast Food Environment

The FFE was measured using data collected from the Food Standards Agency (FSA) website (https://www.food.gov.uk/our-data (accessed weekly between December 2012 to present)). FSA are a governmental department in England that coordinate Local Government inspections of hygiene in shops and services selling fresh food. They publish an open database based on both their internal records, as well as information supplied by Local Governments. Data from the FSA has been collected by the authors from their website since December 2012. Information includes the name of the organisation, address, coordinates, food hygiene rating and a classification of outlet type. We selected two time periods of data closest to the mid-point of survey data collection periods (December 2012 and August 2015). 

Counts of each outlet type per year were aggregated to LSOAs and Local Authority Districts (LADs; town/city-region with mean population size ~180 k). We selected two geographical scales to account for the immediate local neighbourhood context surrounding where individuals reside (LSOAs), as well as the broader context they may live their lives in (LADs). The category ‘takeaway shop’ was used to measure unhealthy outlets within the FFE since these outlets contain fast food outlets and outlets selling nutritionally poor foods for takeaway. We recoded known chain fast food outlets (e.g., MacDonald’s, KFC and Burger King) into this category since they were often recorded as restaurants (identified by name of outlet). We were unable to determine if other outlet types were primarily selling fast food from the data.

We modelled the count of fast food outlets in our analysis to preserve information, rather than identify arbitrary cut points that may produce misleading results [31,32]. We included a sensitivity analysis testing different types of measures to aid comparisons of our analyses to other studies (see Appendix A).

### 2.4. Confounder: Neighbourhood Deprivation

Neighbourhood deprivation was measured using quintiles of the ranks for the Index of Multiple Deprivation (IMD) [33]. IMD is a multidimensional measure of neighbourhood (LSOA) deprivation that is commonly used by national and local governments, as well as by many studies of food environments in the UK [13]. As each constituent country of the UK has their own version of the IMD, which are not comparable, we restricted our analyses to England. 

### 2.5. Additional Measures

We included measures of physical activity and fast food consumption that were only available for wave 6. Physical activity was measured as the number of days per week participants undertook moderate to vigorous physical activity (‘every day’, ‘5–6 days’, ‘3–4 days’ and ‘2 or fewer days’). Fast food consumption was measured as how often a participant consumed fast food (‘weekly’, ‘monthly’ and ‘less/never’). Both of these measures were self-reported in the main questionnaire. Finally, urban areas were also identified using the ONS Urban Rural Classification (2011), which classifies output areas as urban or rural based on population density [34].

### 2.6. Statistical Analysis

Structural Equation Models (SEMs) were used in the analysis. SEMs are a family of multivariate methods which allow the modelling of structural pathways between observed and unobserved variables. The need to make explicit pathways of hypothesised relationships is important. Calls for better conceptual models to identify the pathways through which food environments may influence obesity are not new [21], however they need incorporating within analytical frameworks that allow for their empirical testing [20].

A variety of generalised model specifications and regression analyses were used depending on the outcome variable for a specific pathway. Overweight was modelled using a binomial logit. IMD quintile was analysed using an ordinal regression model. Count of fast food outlets was analysed using a Poisson regression model because (a) the distribution of data for both LSOAs and LADs were right skewed, and (b) negative values of counts were impossible. Alternative model specifications for fast food outlets (e.g., logged outcome and linear model) produced similar findings. Physical activity and fast food consumption were both analysed using an ordinal regression model (binary models did not substantially change the findings). A total of three to five models were run initially testing for observed confounding at each geographical scale (dependent on the number of outcomes), with seven models when introducing fast food consumption and physical activity (which have important implications for the possibility of false positives due to multiple testing). In the sensitivity analyses, change over time measures were modelled as linear OLS due to their normal distributions.

## 3. Results

As shown in Table 1 (*n* = 9736), the prevalence of overweight declined slightly (1%) between waves. Individuals had greater exposure to fast food outlets over time. The largest increase was at the LAD level (34.9 additional outlets) compared to LSOA level (0.13). The distribution for deprivation remains similar across waves, with more participants from the most deprived areas in each wave.

**Hypothesis** **1** **(H1).**
*Individuals who live in areas with more fast food outlets have a higher likelihood of being overweight.*


Figure 1 (insert A and B) presents results from two separate SEMs for the count of fast food outlets at LSOA and LAD level respectively. Full statistical output is provided in the (see Appendix A). For the LSOA count of fast food outlets, there was no association to whether an individual was overweight at both ages. This contrasts to findings when using the count of fast food outlets at the LAD level. There were positive associations to overweight at both age 11/12 (Odds Ratio (OR) = 1.0006, 95% Confidence Intervals (CIs) = 1.0002, 1.0009) and 14/15 (OR = 1.0005, 95% CIs = 1.0002, 1.0008). While effect sizes were small, these represent a one-unit increase in the count of fast food outlets. Since the mean number of fast food outlets at this scale was large (see Table 1), the translation of the actual effect size is reasonable. For example, at age 11/12, a 100-unit difference in exposure would equate to a 5.9% increase in the probability of being overweight (to provide context, for this same time period—median LAD number of takeaways for England was 118 (mean 178), with an interquartile range of 72–239). We accept hypothesis one at the LAD level and reject it at the LSOA level. 

**Hypothesis** **2** **(H2).**
*Individuals who live in deprived neighbourhoods have a higher likelihood of being overweight.*


At both ages, we find evidence of social gradients in the risk of overweight among adolescents (see Figure 1C and Appendix A). At age 11/12, there is a dose-response relationship whereby we detect greater odds of overweight with greater levels of deprivation. Participants who resided in the most deprived quintile had 89% (OR = 1.89, 95% CIs = 1.64, 2.18) higher odds of being overweight compared to those in the least deprived quintile. At age 14/15, we find similar associations albeit not all quintiles were statistically significant. We find a smaller effect size for deprivation with individuals in the most deprived quintile having 41% (OR = 1.41, 95% CIs = 1.16, 1.71) higher odds of being overweight than those in the least deprived quintile. The smaller effect sizes at age 14/15 was due to controlling for overweight status in the previous wave. This confirms hypothesis two.

**Hypothesis** **3** **(H3).**
*Neighbourhood deprivation confounds the association between fast food outlets and likelihood of being overweight.*


Having identified associations between fast food outlets and neighbourhood deprivation to overweight independently, we next investigated whether the association between fast food outlets and overweight is confounded by deprivation (Figure 1D,E and Appendix A). Findings display associations between neighbourhood deprivation to both count of fast food outlets and overweight. No associations were detected between fast food outlets and overweight. For deprivation, the associations with overweight remain like those described in Hypothesis two, suggesting the consistency in evidence for social inequalities in overweight risk. We also detect strong positive associations between deprivation and count of fast food outlets, suggesting greater exposure of adolescents in deprived areas to fast food outlets compared to those in the least deprived quintiles. The associations between LAD density of fast food outlets and overweight for both ages have now disappeared once we accounted for the confounding effect of deprivation (with no association at the LSOA level). Sensitivity analyses testing alternative specifications of the FFE exposure found that associations were consistently non-significant or confounded by deprivation (see Appendix A). Hypothesis three is confirmed. 

**Hypothesis** **4** **(H4).**
*In areas where the number of fast food outlets increased, individuals were not more likely to be overweight.*


In total, 1351 (14.8%) individuals moved LSOA between ages 11/12 and 14/15. We considered for those who moved whether there was an association between the change in the FFE exposure (count of fast food outlets) between waves and overweight at ages 14/15, adjusting for the confounding effect of deprivation at wave 6 (Table 2). We find no associations at either geographical scales (models A and B). Repeating the analyses with the alternative measures of FFE did not alter these findings (results not shown).

Overall change in takeaway count between ages 11/12 and 14/15 for all participants irrespective of whether they moved or not was examined in their association to overweight in wave 6 (Appendix A). We found no association between the change in count and risk of overweight at either geographical scale, as well as evidence that the association is confounded by deprivation (e.g., individuals who resided in deprived areas saw larger increases in the number of fast food outlets between waves). 

Hypothesis four is therefore accepted.

**Hypothesis** **5** **(H5).**
*The confounding role of neighbourhood deprivation persists even after controlling for diet and physical activity.*


Diet and physical activity were next introduced into our SEM models (see Figure 2 and Appendix A). At the LSOA level, the associations for fast food outlets to overweight at both ages remain null and there was no association to consumption of fast food either. At the LAD level, there was no association between density of fast food outlets and overweight at either age. There is a positive association where a greater density of fast food outlets is associated to greater consumption of fast food (OR = 1.0008, 95% CIs = 1.0005, 1.001).

There were social inequalities evident in overweight, fast food consumption and physical activity. Deprivation level at age 14/15 was associated to physical activity, with individuals in the most deprived quintiles more likely to engage in fewer days of exercise in a week. Deprivation level at age 14/15 was also positively associated to fast food consumption, with individuals in the most deprived quintile being more likely to consume fast food at higher frequencies. Adolescents who resided in more deprived areas were at greater odds of being obese at both time periods. Similarly, smaller effect sizes were observed in ages 14/15 than in ages 11/12 due to attenuation following adjustment for previous overweight status in the age 14/15 analysis. 

We find mixed associations for how fast food consumption and physical activity are associated to overweight. Individuals who engaged in physical activity less frequently were more likely to be overweight; participants who undertook two or fewer days of physical activity a week had more than two times the odds (OR = 2.04, 95% CIs = 1.65, 2.51 in both models) of being overweight than those who were physically active every day. For fast food consumption, an association was also detected. Participants who consumed fast food weekly had ~20% lower odds (OR = 0.81, 95% CIs = 0.67, 0.97) of being overweight than compared to those who consumed fast food rarely or never. This has implications for the interpretation of the association between density of fast food outlets and fast food consumption, as there is no clear pathway to overweight. 

The inconsistency of associations between fast food outlets and our outcomes, as well as the lack of a clear pathway between fast food consumption to overweight, leads us to cautiously accept hypothesis five.

## 4. Discussion

### 4.1. Key Findings

Our study utilises a longitudinal design to demonstrate that neighbourhood deprivation confounds the association between the FFE and overweight in adolescents. Associations between the density of fast food outlets surrounding participants and overweight were largely inconsistent across analyses. Detected associations disappeared following accounting for the confounding effect of deprivation, with deprivation strongly associated to both density of fast food outlets and odds of being overweight. We provide a rigorous evaluation of our model and series of sensitivity analyses that demonstrate that our findings are relevant and consistent. This is important given the methodological considerations which may contribute to evidential inconsistency [11,13]. We also showcase how path analysis is a valuable tool for identifying the mechanisms and pathways through how variables are inter-related to understand the role of contextual factors.

### 4.2. Interpretation

Our findings contribute important longitudinal evidence to a largely cross-sectional body of literature [10]. In particular, previous studies employing cross-sectional data are more likely to report that an association exists between FFE and body weight [11], which contrasts to evidence utilising longitudinal data including our study and others [17,18]. Our study suggests through modelling relationships as pathways, we can reveal that neighbourhood deprivation confounds the association between FFE and overweight. It is plausible that cross-sectional designs may be more susceptible to these biases than longitudinal study designs since they cannot separate out the temporal ordering of exposures and outcomes. However, such an explanation is more nuanced than this since longitudinal study designs are still subject to biases (e.g., selection bias, attrition) and do not ultimately identify causal effects alone. Rather, our study demonstrates the need for better quality evidence that extends longitudinal study designs into modelling of mechanisms clearly to truly understand the role of FFEs. 

This study offers tentative evidence that density of fast food outlets at the city/town scale is associated to fast food consumption. However, the lack of a clear pathway to overweight via fast food consumption suggests this finding should be interpreted carefully. More frequent fast food consumption was associated in this study with lower odds of overweight, which does not follow evidence elsewhere [35]. It may be that after controlling for key determinants including deprivation and being overweight in the previous wave, the estimate effect left over is merely a spurious result, especially given the wide confidence intervals and small effect size. These issues may reflect collider bias in how the model is specified. The strong association between overweight status at both ages may distort the real underlying relationship or produce one where it does not exist. Reverse causation may be an important issue, whereby adolescents who are overweight/obese eat less fast food to manage their body weight. The self-reported nature of the fast food consumption measure in MCS may also partly explain the inverse association to overweight, if there was significant under-reporting of consumption habits.

Our findings suggest that strategies aimed at reducing overweight or obesity prevalence in adolescents should focus on tackling the drivers of social inequalities. Socioeconomic context is often described as a fundamental cause of health inequalities [36], due to the powerful role it plays across multiple health behaviours and outcomes. This is especially so for adolescents who are often unable to modify their socioeconomic contexts. Explanations include a lack of material resources for family’s to afford healthy diets, issues of control and power in decision making, stressful lives and the concentration of related harms via syndemics [24]. Strategies focused on tackling the social determinants of adolescent overweight or obesity may have broader knock on effects to other health outcomes too, suggesting interventions are effective. However, our study only considers one part of socioeconomic context in neighbourhood deprivation, and extending our approach to incorporate greater depth here is key (e.g., utilising latent variables to account for multiple factors simultaneously). Similarly, we only consider one element of the food environment and strategies enabling better access to fresh fruit and vegetables (rather than focusing solely on access to fast food) may be important.

Social and spatial inequalities in overweight, obesity and related health outcomes are prevalent in the UK and have gained considerable policy attention [5]. Our results would suggest that Local Government strategies aimed solely at restricting the location of fast food outlets (both overall or clustered around schools) may be ineffective, especially if they are not tackling the underlying social inequalities and household dynamics which are often the driving reasons behind patterns in excess body weight, unhealthy diets or unhealthy environments among adolescents. Policy efforts should therefore focus on tackling levels of deprivation (e.g., poverty alleviation efforts) or mediating their influences (e.g., subsidizing healthy foods in schools or shops) which have been demonstrated to be effective elsewhere [37,38]. 

### 4.3. Limitations

We utilize novel longitudinal data to contribute to a field dominated by cross-sectional research [11,13]. Extending our approach to incorporate a longer time series or a life course perspective including a greater range of ages beyond our specific cohort will be important to improve the generalizability of our findings. A life course approach will also help to assess if particular periods of life are more susceptible to the effects of the FFE and neighbourhood deprivation than others, as well as allowing the consideration of time lag or accumulated effects. Our analyses only consider a three year period in-between waves, which may be too short to identify any changes over time or for contextual effects to impact body weight significantly. While other longitudinal research studying a longer period of adolescence body weight its association to the FFE has found no association [18], it is plausible that these influences may become more important once adolescents move away from home and have more control over their food choices [10,11,12].

We utilize a novel methodological to test specific pathways rather than relying on associational based analyses. This is an important step for future research, both in setting out clearly our assumptions about how phenomena operate, as well as being able to test specific mechanisms rather than treating geographical context as a ‘black box’. The range of pathways and determinants we included is narrow and future research should seek to build more detailed models to assess the role of geographical context. For example, we only consider one feature of the built environment in fast food outlets, however there are other features that have been previously demonstrated to be associated to overweight (e.g., green space, access to fresh fruit and vegetables, and neighbourhood walkability). Our study cannot be dismissive of these features and understanding the extent that neighbourhood deprivation confounds them is a key future research area.

Multi-level SEM models might be helpful for driving this research agenda forward, as they can explicitly account for geographical factors unlike our purely individual-level only analysis. Additionally, we measure the FFE and neighbourhood deprivation using single measures only which are unlikely to capture their full extents. SEMs may offer a solution here through allowing latent variables to capture their broader contexts and accommodate for the complex and interrelated factors within each of these concepts [27]. Thinking about the holistic and relational ways in which contextual factors influence our health is a necessary step for advancing health geography research [21].

Our two measures of geographical context, fast food outlets and neighbourhood deprivation, were also limited and may not accurately measure the issues they seek to describe. Firstly, we calculate the density of fast food outlets based on numbers of chain fast food restaurants and those classified as ‘takeaway shops’. It is possible that those classified as takeaways did not sell fast food. We did not exactly know what was being sold in each outlet (nor was data available on this) and therefore our measure may over-estimate the exposure of participants to fast food outlets. There were no alternative datasets available that provided historical data, a common issue in retrospective health geography research. Previous research has showed that this dataset is a valid measure for food outlets [39]. We also do not consider other aspects of the retail fast food environment that may influence risk of overweight. For example, a high density of fast food outlets may only matter when combined with poor access to healthier foods [2,4]. Extending our work to incorporate more measures of the FFE might be useful for robustly assessing the contribution of the FFE to overweight.

Second, we use a composite index to measure neighbourhood deprivation rather than focusing on specific measurable features of neighbourhood deprivation. While the index of multiple deprivation measure is a multi-dimensional measure covering issues including income, education, health and employment [33], it is a summary measure that may conflate or hide specific issues through ‘averaging them out’. The composite index also contains information on the built environment, that may capture some of the effect of the FFE and leave it in our analyses as a residual effect. Through using quintiles, we may lose out on information within quintiles especially where continuous relationships exist [31], as well as combing areas at the margins of quintiles that otherwise have different socioeconomic contexts into the same groups. 

While we link two longitudinal individual- and geographical-level datasets, our measure of geographical exposure is limited. Future research should look to link data that moves beyond residential location to assess daily movement patterns (e.g., GPS records) that can provide more precise measures of exposure, account for utilisation (rather than just geographical access) or allow the assessment of time sensitive periods in exposure [14]. Identifying the correct context for assessing geographical factors remains an outstanding challenge in health geography [40], in particular for aiding the design of relevant place-based interventions. 

Selection bias may affect the findings from our study. While the MCS is a representative survey, attrition between waves may affect the generalisability of data used in our analyses. Attrition was not randomly distributed throughout our study population. For example, participants from the most deprived quintile and the highest tertile of fast food outlets (LADs) were more likely to have dropped out in-between waves (see p. 9 of Appendix A). This effect was greatest for neighbourhood deprivation. Attrition may have introduced bias into our analyses including affecting our exposure and confounder variables directly (i.e., if individuals more exposed to fast food outlets dropped out, this may lead to associations disappearing). Some of the data we used were self-reported by participants (i.e., physical activity and fast food consumption measures), which may be affected by recall bias or incorrect observations. This issue may be important if the biases introduced by self-reported data are correlated to either our exposure, confounder or outcome variables (e.g., if overweight adolescents are more likely to under-report their fast food consumption, this may lead to an association of fast food consumption being negatively associated to overweight).

Our outcome variable captures body weight as a binary measure of whether an adolescent was overweight or not. While this measure is validated [29], adjusts for age and sex, and is associated with later life obesity and health [8,30], it may introduce bias into our analyses where the binary categorisation artificially hides relationships or through lost information by collapsing continuous information into a smaller number of groups [31]. Changes in our overweight categories may be low over the short study time period (or changes may simply reflect measurement errors), whereas continuous measures might provide additional insights otherwise lost in-between waves. Height, weight and standardised BMI values were not provided due to statistical disclosure issues, limiting our ability to test how much this matters. Future research should look to replicate our analyses with more diverse measures of body weight. We note that other studies exploring the FFE have used binary outcome measures, and where continuous measures of body weight were used they do not always find significant associations [11,12,18], which may suggest that the impact of this limitation may be minor. 

Finally, there are numerous other confounding factors identified in the literatures for both the food environment and determinants of overweight that we have not considered (e.g., access to fresh fruit and vegetables, fast food advertisement exposure, price of foods and income). Future research should examine integrating into our modelling framework to identify how their specific pathways change our observations. Our study helps to showcase how we can use path analysis techniques to model specific pathways and clearly outline our assumptions, which is key for understanding the mechanisms affecting body weight.

## 5. Conclusions

We find evidence that the association between the food environment and overweight in adolescents is confounded by neighbourhood deprivation. Our findings have a wide range of policy, applied, conceptual and methodological applications to researchers. Accounting for the confounding role of deprivation when assessing the nature of geographical contexts on health is key for future research studying social and spatial inequalities. Understanding the underlying systematic pathways which produce such a disadvantage is necessary to design effective interventions. 

## Figures and Tables

**Figure 1 ijerph-18-13212-f001:**
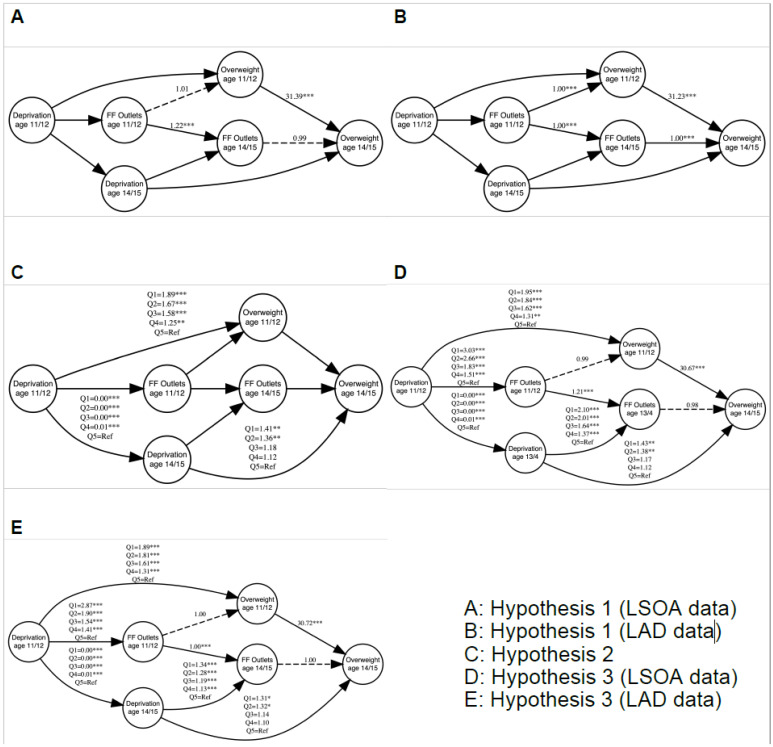
Five Structural Equation Models exploring the extent that the association between density of takeaways and overweight in children is confounded by deprivation. (Note: Odds Ratios are presented. Error terms are not presented to aid visual interpretation. The full outputs for models can be seen in the Appendix A. Dotted lines represent insignificant associations, hard lines represent significant associations. * *p* < 0.05, ** *p* < 0.01, *** *p* < 0.001. LSOA = Lower Super Output Area. LAD = Local Authority District).

**Figure 2 ijerph-18-13212-f002:**
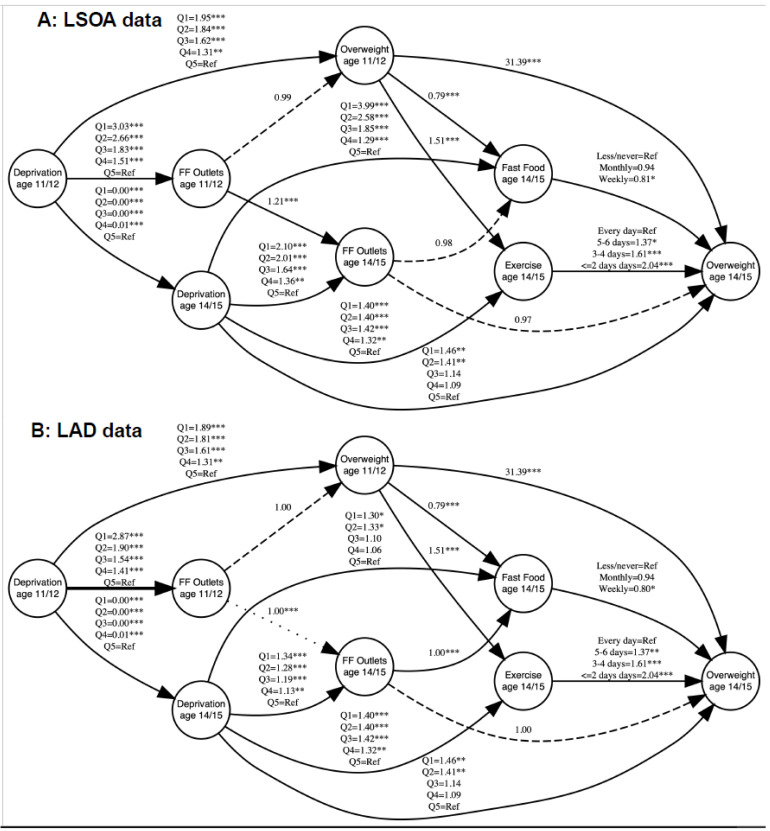
Two Structural Equation Models exploring whether the association between density of takeaways and overweight in children persists after account for fast food consumption and physical activity. (Error terms are not presented to aid visual interpretation. The full outputs for models can be seen in the Appendix A. Dotted lines represent insignificant associations, hard lines represent significant associations. * *p* < 0.05, ** *p* < 0.01, *** *p* < 0.001. LSOA = Lower Super Output Area. LAD = Local Authority District).

**Table 1 ijerph-18-13212-t001:** Analytical sample characteristics by wave.

Measure	Ages 11/12—Wave 5	Ages 14/15—Waves 6
Mean Age	10.6	13.8
Males	49.4%	49.4%
Females	50.7%	50.6%
Overweight	26.4%	25.4%
Mean Fast food outlets in Lower Super Output Area	1.1	1.2
Mean Fast food outlets in Local Authority	179.2	214.1
Deprivation Quintile 1 (Most Deprived)	23.9%	23.7%
Deprivation Quintile 2	19.3%	19.2%
Deprivation Quintile 3	18.8%	18.7%
Deprivation Quintile 4	18.4%	18.6%
Deprivation Quintile 5 (Least Deprived)	19.7%	19.9%

**Table 2 ijerph-18-13212-t002:** Results from a logistic regression examining the association between change in number of fast food outlets and wave 6 overweight (adjusted for neighbourhood deprivation) in participants who migrated (*n* = 1351). Note: CI = 95% Confidence Intervals.

	Odds Ratio	Lower CI	Upper CI	*p* Value
**Model A: Lower Super Output Areas**				
Change in fast food outlets	0.967	0.929	1.007	0.105
**Model B: Local Authority District**				
Change in fast food outlets	1.0001	0.999	1.002	0.903

## Data Availability

Food Standards Agency data are openly available from https://www.food.gov.uk/our-data (accessed weekly between December 2012 to present), with historical data sets freely available via the CDRC (see http://data.cdrc.ac.uk accessed weekly between December 2012 to present). Millennium Cohort Study data are freely available via the UK Data Archive:University of London, Institute of Education, Centre for Longitudinal Studies. (2019). *Millennium Cohort Study: Sixth Survey, 2015*. [data collection]. *4th Edition.* UK Data Service. SN: 8156, http://doi.org/10.5255/UKDA-SN-8156-4 (accessed on 2 August 2019). University of London, Institute of Education, Centre for Longitudinal Studies. (2017). Millennium Cohort Study: Geographical Identifiers, Sixth Survey, 2011 Census Boundaries: Secure Access. [data collection]. UK Data Service. SN: 8232, http://doi.org/10.5255/UKDA-SN-8232-1 (accessed on 2 August 2019). University of London, Institute of Education, Centre for Longitudinal Studies. (2017). Millennium Cohort Study: Fifth Survey, 2012. [data collection]. 4th Edition. UK Data Service. SN: 7464, http://doi.org/10.5255/UKDA-SN-7464-4 (accessed on 2 August 2019). University of London, Institute of Education, Centre for Longitudinal Studies. (2017). Millennium Cohort Study: Geographical Identifiers, Fifth Survey, 2011 Census Boundaries: Secure Access. [data collection]. 2nd Edition. UK Data Service. SN: 7763, http://doi.org/10.5255/UKDA-SN-7763-2 (accessed on 2 August 2019).

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
