# Peer review of "The Association between Fast Food Outlets and Overweight in Adolescents Is Confounded by Neighbourhood Deprivation: A Longitudinal Analysis of the Millennium Cohort Study"

_ijerph, 2021, doi:10.3390/ijerph182413212_

Round 1

Reviewer 1 Report

Congratulations on an excellent paper around an important area. There are some minor spelling and grammar errors but overall there are no suggested changes from this reviewer. However I wonder if there may be more emphasis on the role that social deprivation has upon health and the call for policy change.       

Author Response

Reviewer 1

Comment: Congratulations on an excellent paper around an important area. There are some minor spelling and grammar errors but overall there are no suggested changes from this reviewer. However I wonder if there may be more emphasis on the role that social deprivation has upon health and the call for policy change.    

Response: Thank you for your kind comments. In response we have:

  • We have gone through the paper and made some changes to improve the spelling and grammar throughout (e.g., p3, p9, p10).
  • We agree about mentioning the wider role social deprivation has on health in the policy section. We have added: “Strategies focused on tackling the social determinants of adolescent overweight or obesity may have broader knock on effects to other health outcomes too, suggesting interventions are valuable.” (p10)

Reviewer 2 Report

In attached file

Author Response

Reviewer 2

Comment: The authors deal with an interesting topic in their study. However, for its publication I think it is necessary to consider a number of important considerations. There are some results that I am seriously concerned about being published, and I believe that the entire methodology must be reviewed more thoroughly, since the bias of the study can be important, especially when the authors themselves say that they want to make a special impact on the bias of the studies.

Response: We thank the reviewer for their critical comments and suggestions for how to refine the paper. We hope that the revisions we have made have sufficiently addressed your comments. They certainly have helped us improve the quality of our paper.

Abstract

Comment: It would improve if it were structured by sections.

Response: The IJERPH instructions for authors states “The abstract should be a single paragraph and should follow the style of structured abstracts, but without headings". We have followed the format, but leave the structured sections out as you suggest to match their required format.

Comment: Some quantitative data should be provided in the summary of results.

Response: It was rather difficult to synthesise all our models into the three sentences, given there are many associations and the word count is 200 words. We have decided to not include any and therefore respectfully decline the suggestion. We note that it does not change the narrative too much at least.

Introduction

Comment: The authors aim to establish the extent to which the association between the fast-food environment and being overweight is confused with neighborhood deprivation. Next, the authors clarify which measure they use for the density of fast-food outlets, but do not clarify how they measure the variable "neighborhood deprivation." I think it is necessary to clarify this aspect here too, although it is also clarified in 'materials and methods'.

Response: We agree that mention of the specific variables used in the analyses is distracting at this point in the introduction, especially as they are detailed in greater depth in the methodology. As such, we have removed the statement about using density of fast food outlets here (see p4).

Materials and methods

Comment: Review the abbreviations used, as some are not previously quoted in full text in the document ("LSOA codes"). The authors clarify it later, but not the first time they use it.

Response: We have made the correction suggested and spelled out LSOA in full at the first occurrence (p5).

Comment: Why were the "Z-score values" not available for analysis? How can this affect the results of the study? Clarify.

Response: Data were unfortunately not provided by the MCS team in the data extract we were given. We have added the following statement to clarify the impact this might have: “While the use of a categorical variable is limited through hiding variations within groups, z-score body weight values were not available for analysis.” (p5)

Comment: ¿Are weight and height also self-reported?

Response: They were not and this is noted in the methodology: “Interviewers objectively measured anthropometrics including height and weight of participants” (p5)

Comment: It is unclear how physical activity has been measured. With questionnaire? If so, which questionnaire has been used?

Response: The data were included in the main survey. To clarify this “Both of these measures were self-reported in the main questionnaire.” (p6)

Results

Comment: In the version of the article that I have, the tables do not contain their "table footer" in which the abbreviations contained in the table must be recorded, as well as other necessary clarifications. Nor in this version can I see the figures to be able to evaluate them.

Response: All files, figures and captions were uploaded correctly and we are sorry that you had trouble accessing them. We have updated all Tables now so abbreviations are either spelled out in full or defined.

Discussion

Comment: The authors in the discussion emphasize that mainly in their study they want to highlight the bias that not only cross-sectional studies can have, but also longitudinal ones. Since your study is longitudinal, what biases does it have? Can we then conclude that given the biases that your study may have, the results reached by the authors are erroneous? Or can we conclude that there is really no association between fast food environments and overweight/obesity, and what really influences is the disadvantaged socioeconomic status of the participants? Could it not be that there is a summative effect between variables? Please clarify all these aspects, as they lead to a lot of confusion for the reader.

Response: We agree with your suggestion and have tried to tone down parts of the narrative now to follow your comments. Specifically, we have added the following statements in the discussion now:

  • “Our study suggests that neighbourhood deprivation confounds the association between FFE and overweight, and it is plausible that cross-sectional designs may be more susceptible to these biases than longitudinal study designs. However, such an explanation is more nuanced than this since longitudinal study designs are still subject to biases (e.g., selection bias, attrition) and do not ultimately identify causal effects alone. Rather, our study demonstrates the need for better quality evidence that extends longitudinal study designs to truly understand the role of FFEs and places a critical eye over any confounding processes.” (p9)
  • “Selection bias may affect the findings from our study. While the MCS is a representative survey, attrition between waves may affect the generalisability of data used in our analyses. Some of the data we used were self-reported by participants, which may be affected by recall bias or incorrect observations.” (p11)

Comment: I am very concerned about some of the results presented by the authors, such as the following collected in the discussion: "The most frequent consumption of fast food was associated in this study with a lower probability of being overweight." This aspect must be analyzed and discussed by the authors before being published in much more depth and contrast all these aspects much more thoroughly with the published literature. I believe that a self-reported measure of consumption is not sufficient reason for this great bias, and if this measure includes this important bias, it should be reconsidered to publish this study.

Response: We share similar concerns, as we also running the original analyses. We have gone back to the initial data to re-check our analyses. They are robust and not due to statistical oversight. Indeed, running an unadjusted logistic regression model of just fast food consumption on overweight/obesity as our outcome presents a similar association. The percentage of overweight/obesity in people who consumed fast food weekly was 24%, compared to 26% in those less than monthly/never – so there was little difference between the groups. We also tried re-running analyses for an associated outcome (body fat percentage, available only for wave 6) and the results were consistent.

We have edited the paragraph discussing this finding in the discussion to try and add some caveats to the explanation and share your concerns more explicitly.. Specifically, we have added more references here too, where before we have no citations. It now reads:

“This study offers tentative evidence that density of fast food outlets at the city/town scale is associated to fast food consumption. However, the lack of a clear pathway to overweight via fast food consumption suggests this finding should be interpreted carefully. More frequent fast food consumption was associated in this study with lower likelihood of overweight, which does not follow evidence elsewhere [34]. It may be that after controlling for key determinants including deprivation and being overweight in previous wave, that the effect left over is merely a spurious result, especially given the wide confidence intervals. Reverse causation may be an important issue, whereby adolescents who are overweight/obese eat less fast food to manage their body weight. The self-reported nature of the fast food consumption measure in MCS may also partly explain the inverse association to overweight, if there was significant under-reporting of consumption habits.” (p10)

We also note here for your reference, but in the paper as it doesn’t really fit, that similar findings have been noted previously in international comparisons e.g., https://pubmed.ncbi.nlm.nih.gov/25488096/ (this is, however, less relevant to our study).

Comment: It seems clear that the issue of social inequalities must be addressed as a strategy for the prevention of overweight/obesity, but I believe that the authors should also focus on the other predisposing factor that has been amply demonstrated in the scientific literature, which is the ease of access to fast food. Please clarify all these aspects, because reading the article gives the feeling that the authors downplay this factor.

Response: The key finding of our paper was that there were no associations between the fast food environment and overweight/obesity, with socioeconomic status explaining any association as a confounding factor. This is why we focus on tackling deprivation as the main policy issue and not access to fast food. We feel it would lessen our narrative to say fast food environment doesn’t matter, but then talk about it as an important policy issue. We wanted to acknowledge a broader point you have raised, in that other features of food environments might still be useful policy options. As such, we have added the following: “Similarly, we only consider one element of the food environment and strategies enabling better access to fresh fruit and vegetables (rather than focusing solely on access to fast food) may be important.” (p10)

Comment: In the section "strengths and limitations", I consider that the authors do not sufficiently detail the limitations of their study, such as the fact that all measures are self-reported, which should be treated in much more detail in terms of the bias that this may introduce in the study. Also, in this section the authors should deal in more depth with the appropriateness or not that the measure of "privacy situation used" is.

Response: We have added more details to the limitations section to cover your helpful suggestions in acknowledging our limitations as concisely as possible. We added: “Selection bias may affect the findings from our study. While the MCS is a representative survey, attrition between waves may affect the generalisability of data used in our analyses. Some of the data we used were self-reported by participants, which may be affected by recall bias or incorrect observations.” (p10-11)

We were unsure what you meant by “measure of privacy situation used”. We do not mention about privacy anywhere in the paper, nor have any measures related to this in our analyses. All data are collected externally by the Millennium Cohort Survey team who have appropriate consent procedures in place and shared to researchers as secondary data (accessed by application by accredited researchers). We do not feel we need to add any more details in here therefore.

Reviewer 3 Report

This article of Green and colleagues presents the role of the neighbourhood deprivation in association between fast food outlets and overweight in adolescents on national level.

I appreciate the opportunity to read and review this interesting and well prepared and written article. The authors have applied an appropriate and strong methodology, correct presentation and interpretation of results, which can help the policymakers in better understanding and tackling the social determinants of overweight and obesity in adolescents and in later life.

Please give the citation of Food Standards Agency (FSA) website in the Methodology part 2.3. and for the ONS Urban Rural Classification in part 2.5.

Abbreviations used in tables 1 (LSOAs , LAD, IMD Q1-Q5) should be described in the table legend.

Calculations of OR should be represented by the percentage - 95% or 99%, of confidence level (e.g. OR = 1.89, CIs 95% = 1.64,2.18).

CI must be represented in the same way:  OR = 1.89, CI = 1.64,2.18; or OR = 0.81, CIs = 0.67-0.97).

Author Response

Reviewer 3

Comment: This article of Green and colleagues presents the role of the neighbourhood deprivation in association between fast food outlets and overweight in adolescents on national level. I appreciate the opportunity to read and review this interesting and well prepared and written article. The authors have applied an appropriate and strong methodology, correct presentation and interpretation of results, which can help the policymakers in better understanding and tackling the social determinants of overweight and obesity in adolescents and in later life.

Response: We would like to thank you for your kind and positive comments about our paper.

Comment: Please give the citation of Food Standards Agency (FSA) website in the Methodology part 2.3. and for the ONS Urban Rural Classification in part 2.5.

Response: We have added references for both of these data now (see pp5-6)

Comment: Abbreviations used in tables 1 (LSOAs , LAD, IMD Q1-Q5) should be described in the table legend.

Response: We have updated all the tables now to spell out abbreviations or define acronyms.

Comment: Calculations of OR should be represented by the percentage - 95% or 99%, of confidence level (e.g. OR = 1.89, CIs 95% = 1.64,2.18).

Response: We have added ‘95%’ to each mention of the confidence intervals in the text now.

Comment: CI must be represented in the same way:  OR = 1.89, CI = 1.64,2.18; or OR = 0.81, CIs = 0.67-0.97).

Response: We have revised all mentions in text to report consistently (selecting commas).

Round 2

Reviewer 2 Report

I believe that the modifications made by the authors are not sufficient to respond to the aspects that I expressed in my comments and that seriously concern me

Author Response

Reviewer 2

Comment: I believe that the modifications made by the authors are not sufficient to respond to the aspects that I expressed in my comments and that seriously concern me

Response: We have now revised our paper and added more details in line with the limitations that you originally identified. Indeed the limitations section has now increased from one paragraph to two pages of materials. Specifically, we have added the following:

  • We have added quantitative data into the abstract to describe the associations that we detect (see abstract)
  • We have added more clarity that z-score BMI (including height and weight) were simply not provided by the data controllers (p5)
  • We have significantly extended the limitations section of our paper – it has increased from one paragraph that was < half a page, to 2 full pages of text (see Section 4.2, p11-13). We now cover the following themes which talk about the issues with bias that you previously identified in far greater detail: (i) impact of short time period may not be long enough to detect effects, (ii) need to consider broader contextual factors that might explain why the fast food environment matters, (iii) need to move away from use of single variables that may not fully capture associations, (iv) bias introduced by inaccurate geographical measures for both fast food outlets and neighbourhood deprivation (including how each are defined may introduce bias), (v) need to have better measures of geographical exposure, (vi) selection bias due to attrition and response bias from self-reported data, (vii) bias introduced by using a binary outcome variable, and (viii) need to account for other confounders.
  • We have added an additional explanation for the inverse association between fast food consumption and overweight in the discussion (potential collider bias – p10) and also note in the limitations section over how response bias in the self-reported questions may lead to a biased association (p12).

We accept that no study is perfect and hope that the detailed limitations section can go a long way to helping to acknowledge your concerns, while also stating how our paper is scientifically robust.